# Evaluation of Antibody Kinetics Following COVID-19 Vaccination in Greek SARS-CoV-2 Infected and Naïve Healthcare Workers

**DOI:** 10.3390/jpm13060910

**Published:** 2023-05-29

**Authors:** George Pavlidis, Vasileios Giannoulis, Maria Pirounaki, Ioannis C. Lampropoulos, Eirini Siafi, Alkippi Nitsa, Efthymia Pavlou, Anna Xanthaki, Garyfallia Perlepe, Sotirios P. Fortis, George Charalambous, Christos F. Kampolis, Ioannis Pantazopoulos

**Affiliations:** 1Department of Emergency Medicine, Hippokration General Hospital, 11527 Athens, Greece; eirinisiafi@gmail.com (E.S.); drcharalambous@yahoo.gr (G.C.); chkamp77@gmail.com (C.F.K.); 2Transfusion and Haemophilia Centre, Hippokration General Hospital, 11527 Athens, Greece; giavasilio@gmail.com (V.G.); pavlou@hippocratio.gr (E.P.); 3Second Department of Internal Medicine, National and Kapodistrian University of Athens, School of Medicine, Hippokration General Hospital, 11527 Athens, Greece; pirounakimaria63@gmail.com; 4Respiratory Medicine Department, Faculty of Medicine, University of Thessaly, 41500 Larissa, Greece; i.ch.lampropoulos@gmail.com (I.C.L.); perlepef19@gmail.com (G.P.); pantazopoulosioannis@yahoo.com (I.P.); 5Microbiology Department, Hippokration General Hospital, 11527 Athens, Greece; alkippi@yahoo.gr (A.N.); annaxanth61@gmail.com (A.X.); 6Laboratory of Reliability and Quality Control in Laboratory Hematology (HemQcR), Department of Biomedical Sciences, School of Health and Caring Sciences, University of West Attica, 12243 Egaleo, Greece; sfortis@uniwa.gr; 7Department of Emergency Medicine, Faculty of Medicine, University of Thessaly, 41500 Larissa, Greece

**Keywords:** severe acute respiratory syndrome coronavirus 2, coronavirus disease 2019, vaccine, neutralizing antibodies, immunity, healthcare workers

## Abstract

We investigated the antibody kinetics after vaccination against COVID-19 in healthcare workers of a Greek tertiary hospital. Eight hundred and three subjects were included, of whom 758 (94.4%) received the BNT162b2 vaccine (Pfizer-BioNTech), eight (1%) mRNA-1273 (Moderna), 14 (1.7%) ChAdOx1 (Oxford-AstraZeneca) and 23 (2.9%) Ad26.COV2.S (Janssen). Before the second dose, at 2, 6 and 9 months after the second dose and at 2 and 6 months after the third dose, anti-spike IgG were quantified by the chemiluminescence microparticle immunoassay method. One hundred subjects were infected before vaccination (group A), 335 were infected after receiving at least one vaccine dose (group B), while 368 had never been infected (group C). Group A presented a greater number of hospitalizations and reinfections compared to group B (*p* < 0.05). By multivariate analysis, younger age was associated with an increased risk of reinfection (odds ratio: 0.956, *p* = 0.004). All subjects showed the highest antibody titers at 2 months after the second and third dose. Group A showed higher antibody titers pre-second dose, which remained elevated 6 months post-second dose compared to groups B and C (*p* < 0.05). Pre-vaccine infection leads to rapid development of high antibody titer and a slower decline. Vaccination is associated with fewer hospitalizations and fewer reinfections.

## 1. Introduction

Severe acute respiratory syndrome coronavirus 2 (SARS-CoV-2) causes coronavirus disease 2019 (COVID-19), which constitutes a major global health threat in recent years. Numerous effective and safe vaccines have been developed in order to reduce the risk of SARS-CoV-2 infection, severe disease and death, including mRNA [BNT162b2 (Pfizer-BioNTech) and mRNA-1273 (Moderna)] and adenovirus viral vector [ChAdOx1 (Oxford-AstraZeneca) and Ad26.COV2.S (Janssen)] vaccines [1,2,3,4]. SARS-CoV-2 contains the spike (S) glycoprotein on its surface, which can be recognized by the immune system, causing a protective humoral response, the main target of current vaccines. The spike subunit 1 holds the receptor-binding domain (RBD) that mediates viral binding to angiotensin-converting enzyme-2 (ACE-2) receptor on the host cells, whereas spike subunit 2 mediates the fusion between viral and cellular membranes [5]. The vaccines induce the production of spike (S) glycoprotein-specific immunoglobulin G (IgG) against SARS-CoV-2, which can effectively neutralize the infection. Therefore, serology-based assays have been used to detect spike protein domain antibodies induced either by previous viral exposure or by vaccination [6]. Clinical trials have reported that mRNA vaccines efficacy was approximately 95% against SARS-CoV-2 infection, while viral vector vaccines showed approximately 70–75% efficacy [7]. A recent systematic review and meta-analysis of 28 randomized controlled trials revealed that the efficacy of SARS-CoV-2 vaccines is higher for preventing severe infection, hospitalization and death than for preventing milder infection [8]. Among previously infected subjects who have completed a primary series of any COVID-19 vaccine, vaccines remain effective against SARS-CoV-2 reinfection during periods of Alpha, Delta, and Omicron variants, with an efficacy ranging from 60% to 94% over at least 9 months post-vaccine [9].

The duration of vaccination-induced immunity against SARS-CoV-2 remains unclear. Several studies have shown a significant decline in antibody titers three and six months after mRNA-based vaccination in subjects who received two doses [10,11]. Thus, the booster third dose applied six months after completing the two-dose regimen increased substantially antibody concentrations [12]. On the contrary, other studies have reported that humoral response was well detectable from 6–10 months after the second dose of mRNA vaccine [13,14,15]. Additionally, antigen exposure from natural SARS-CoV-2 infection, either before or after vaccination, has been demonstrated to importantly augment the potency and breadth of immune responses [16]. Ongoing somatic mutations of antibody genes, memory B cell clonal turnover and production of monoclonal antibodies which are highly resistant to RBD mutations, including those found in circulating SARS-CoV-2 variants, are among the mechanisms proposed for these robust serological responses observed in convalescent individuals. Furthermore, B cell clones potentially serve as an immune reservoir and may expand significantly after vaccination [17].

The aim of the present real-world study was to investigate the kinetics of anti-S-RBD IgG antibodies after vaccination of Greek healthcare workers over a period of up to 6 months post-third vaccine dose. Antibody kinetics were also compared between convalescent SARS-CoV-2 infected and naïve individuals. In addition, the potential influence of clinical characteristics and co-morbidities on humoral response to COVID-19 vaccine, as well as the possible association of the time from vaccine to SARS-CoV-2 infection or reinfection, were investigated.

## 2. Materials and Methods

### 2.1. Study Design

Eight hundred and twenty-four healthcare workers from the Hippokration General Hospital in Athens, Greece, were evaluated for eligibility in the present retrospective observational study estimating the antibody kinetics after COVID-19 vaccination. Inclusion criteria for participation in this study were age > 18 years and eligibility for vaccination according to the National Immunization Program for COVID-19. Exclusion criteria were hematological malignancies, active solid malignancies, immunosuppressive treatment (long-term therapy with high doses of corticosteroids, chemotherapy, or biological regimen) and end-stage renal or liver disease. Of the 824 enrolled subjects, 21 individuals were excluded from the study due to unwillingness to participate (*n* = 10), history of hematological malignancies (*n* = 2), history of active malignancies (*n* = 4), or being in receipt of immunosuppressive therapy (*n* = 5). Hence, 803 subjects [540 females with a median age of 51 (interquartile range: 40–58) years and 263 males with a median age of 50.5 (interquartile range: 40–56) years] were included in the study. Of those, 100 had been previously infected with SARS-CoV-2 as determined by a positive nucleic acid real-time polymerase chain reaction (RT-PCR) assay. Seven hundred fifty-eight subjects (94.4%) received the BNT162b2 vaccine [Pfizer-BioNTech; of those, 84 subjects received only 2 doses of the BNT162b2 vaccine, 2 subjects received as a booster second dose the Ad26.COV2.S vaccine (Janssen), 662 received a booster third dose of BNT162b2 vaccine, 8 received as a booster third dose the ChAdOx1 vaccine (Oxford-AstraZeneca), and 2 received as a third dose Ad26.COV2.S], 8 (1%) the mRNA-1273 (Moderna; 2 subjects received only 2 doses of the mRNA-1273 vaccine, 3 received 3 doses of mRNA-1273, and 3 received a booster third dose of the BNT162b2 vaccine), 14 (1.7%) ChAdOx1 (one subject received only 2 doses of the ChAdOx1 vaccine, 11 received a booster third dose of the BNT162b2 vaccine, and 2 received a third dose of mRNA-1273) and 23 (2.9%) Ad26.COV2.S (11 of those received 2 doses of Ad26.COV2.S, 11 subjects received a second dose of BNT162b2 vaccine, and 1 subject received a second dose of mRNA-1273) in the period between January 2021 and April 2022. The timeline of vaccine administration in relation to history of SARS-CoV-2 infection in the study population is illustrated in Figure 1. Participants were followed up for a further 6 months after the booster dose to evaluate whether SARS-CoV-2 infection occurred.

A form containing demographics, clinical characteristics, and SARS-CoV-2 infection before and after vaccinations was completed by all participants.

Blood samples were taken for serological assessment at: time 1 (T1), approximately 2 months (mean time 8.3 ± 6.1 weeks) after the first dose of BNT162b2 (Pfizer/BioNTech), mRNA-1273 (Moderna), ChAdOx1 (AstraZeneca) and Ad26.COV2.S (Janssen) vaccine and before the second dose of BNT162b2 (Pfizer/BioNTech), mRNA-1273 (Moderna) and ChAdOx1 (AstraZeneca); time 2 (T2), 2 months (mean time 8.1 ± 6.8 weeks) after the second dose of the aforementioned vaccines; time 3 (T3), 6 months (mean time 23.6 ± 10.6 weeks) after the second dose; time 4 (T4), 9 months (mean time 35.9 ± 6.4 weeks) after the second dose [before the third dose of BNT162b2 (Pfizer/BioNTech) and mRNA-1273 (Moderna) vaccines]; time 5 (T5), 2 months (mean time 7.9 ± 6.3 weeks) after the third dose of the aforementioned vaccines; time 6 (T6) 6 months (23.6 ± 8.1 weeks) after the third dose. Sampling time points differed among SARS-CoV-2 infected subjects due to modification of vaccine schedule by the infection.

Participant data, including name, medical history, and antibody titers, were kept confidential in accordance with the principles of the General Protection Regulation.

### 2.2. Measurements of Antibodies

Serum was separated and processed within 4 hours of blood collection. All analytical procedures were performed at the Department of Microbiology, Hippokration General Hospital, Athens, Greece.

Anti-S-RBD IgG antibodies, which indicate a humoral response to either a previous SARS-CoV-2 infection or a vaccine, were quantified by the chemiluminescence microparticle immunoassay (CMIA) SARS-CoV-2 IgG II Quant on the Alinity i system (Abbott Diagnostics, Chicago, IL, USA). The method has a sensitivity of 98.3% and a specificity of 99.4%, as previously published [18]. The analytical measurement range was stated as 21 to 40,000 AU/mL, while the reportable range was from 6.8 to 80,000 AU/mL. Antibody titers ≥ 50 AU/mL were considered positive, as per manufacturer recommendations [18]. 

### 2.3. Statistical Analysis

All analyses were performed using the Statistical Package for Social Sciences (IBM SPSS Statistics for Windows, version 26.0. Armonk, NY, USA: IBM Corp.) and GraphPad Prism version 9.3.1 for Windows (GraphPad Software, San Diego, CA, USA, www.graphpad.com, accessed on 5 May 2023). Categorical variables are presented as percentages of the study population and were compared using the chi-squared test. Continuous data are expressed as median with interquartile range (first quartile–third quartile) or mean ± standard deviation and were compared using an independent sample *t*-test. To find the antibody levels in the study groups, one-way analysis of variance (ANOVA) was applied between the three study groups (pre-vaccine SARS-CoV-2 infected subjects, post-vaccine SARS-CoV-2 infected subjects and SARS-CoV-2 non-infected subjects). Univariate and multivariable analyses were performed to find the association of the time from the second vaccine dose to the first SARS-CoV-2 infection and the time from the second vaccine dose to SARS-CoV-2 reinfection with co-morbidities. Logistic regression was also applied to investigate the relationship between SARS-CoV-2 reinfection and co-morbidities, and their corresponding odds ratio (OR) and 95% confidence interval (CI) were assessed. A *p*-value < 0.05 was considered statistically significant.

## 3. Results

The demographic and clinical characteristics of the study population are presented in Table 1. A total of 100 subjects were infected with SARS-CoV-2 before vaccination (group A), 335 were infected after receiving at least one dose of vaccine (group B), while 368 were not infected (group C). Subjects in group C were older compared to subjects in groups A and B (median age 53 vs. 46.5 vs. 49 years, *p* < 0.001). There were no differences among the three groups in terms of gender, as well in the type of the vaccine, with the exception of the percentage of subjects who received the Ad26.COV2.S vaccine between groups A and C (6% vs. 1.9%, *p* = 0.027; Table 1). Individuals in group A presented a greater rate of hospitalizations (8% vs. 1.5%, *p* = 0.001) and reinfections (28% vs. 3.3%, *p* < 0.001) compared with individuals in group B. No gender differences were observed among subjects who were hospitalized (*p* = 0.102). In group B, the time from the second vaccine dose to the first SARS-CoV-2 infection was 11.9 [interquartile range (IQR): 8.2–14] months. The time from the second vaccine dose to SARS-CoV-2 reinfection was shorter in group A compared with group B (median time 8.9 vs. 11.4 months, *p* = 0.014). Subjects in group A showed a greater rate of coronary artery disease (CAD) (5% vs. 1.5% vs. 1.1%, *p* = 0.027) and of immunosuppression (3% vs. 0.6% vs. 0.5%, *p* = 0.049) compared with groups B and C. Besides b-blocker utilization, there were no significant differences in medication, as well as in the number of co-morbidities among the three study groups (Table 1). 

Among pre-vaccine SARS-CoV-2-infected healthcare workers (*n* = 100), there were 72 (72%) subjects with one confirmed infection and 28 (28%) that had been infected twice (four subjects were reinfected pre-vaccination and 24 after at least one vaccine dose). Additionally, the majority (73%) of post-vaccine SARS-CoV-2 infected subjects were infected after the booster third dose. In terms of total SARS-CoV-2 reinfection (*n* = 39), more than half of the subjects (54%) were reinfected after the booster third dose (Table 2). Clinical risk factors and co-morbidities were not correlated with the time from the second vaccine dose to SARS-CoV-2 infection or reinfection, whereas age was associated with the time from the second dose to reinfection (Appendix A). By multivariable logistic regression analysis, younger age was associated with an increased risk of SARS-CoV-2 reinfection (OR = 0.956; 95% CI: 0.927–0.986; *p* = 0.004) in a model including sex, obesity, hypertension, dyslipidemia, diabetes mellitus and CAD (Appendix A).

All study subjects showed the highest IgG titer at 2 months after the second and third doses. Subjects in group A showed higher antibody titers before the second dose (T1) [7181 (IQR: 1120–19.257) vs. 1136 (IQR: 330–2859) vs. 623 (IQR: 113–2190) AU/mL, *p* = 0.009] and remained at higher levels at 2 (T2) [10.134 (IQR: 4852–20.025) vs. 7144 (IQR: 2916–13,848) vs. 6169 (2399–12,554) AU/mL, *p* < 0.001] and 6 months after the second dose (T3) [7876 (IQR: 1792–12,944) vs. 1426 (661–2968) vs. 1266 (613–2690) AU/mL, *p* = 0.045] compared with groups B and C. There were no statistically significant differences in antibody titers among the three study groups at 9 months post-second vaccine dose (T4) and at 2 months post-third dose (T5) (*p* = 0.686 and *p* = 0.231, respectively). Finally, subjects in group B exhibited higher antibody titers at 6 months post-third dose (T6) [23,848 (IQR: 9829–42,075) vs. 13,590 (7302–17,994) vs. 8882 (IQR: 2474–17,819) AU/mL, *p* = 0.007] compared with groups A and C (Table 3; Figure 2). In groups A and B, a not significant increase in antibody titers was observed between T5 and T6 time points (*p* = 0.177 and *p* = 0.098, respectively) while subjects in group C presented a significant decrease in antibody levels in the same time period (*p* = 0.043).

In the whole study population, the kinetics of antibody responses were not affected by co-morbidities. Furthermore, there were no differences between men and women regarding the kinetics of antibodies within each sampling time (*p* > 0.05 for all comparisons; data not shown). Regarding age, the youngest group (19–40 years) presented higher antibody titers at 2 months after the second dose (T2) compared with the 41–51, 52–57, and 58–70 age groups (*p* = 0.005, *p* = 0.004, and *p* < 0.001, respectively) and at 2 months after the third dose (T5) compared to the eldest group (58–70 years) (*p* = 0.036). Moreover, the eldest group had higher antibody levels compared with 41–51 and 52–57 age groups (*p* = 0.045 and *p* = 0.026, respectively) at 9 months after the second dose (T4) and lower antibody levels compared with 52–57 age group 2 months after the booster third dose (T5, *p* = 0.045; Figure 3).

## 4. Discussion

In the present study, we monitored antibody responses from the 2 months after the first vaccine dose up to 6 months after the booster third dose in a cohort of 803 Greek healthcare workers, the majority of which had received mRNA-based vaccines. To the best of our knowledge, this is the first real-world study in Greece investigating antibody kinetics after vaccination in a large cohort of both SARS-CoV-2 infected and naïve individuals. In contrast to corresponding serological studies involving general population or special patient groups, our study evaluated the humoral immune responses to vaccination in healthy adults or adults with chronic mild-to-moderate comorbid conditions who were at increased risk of SARS-CoV-2 infection due to professional exposure.

We found that anti-S-RBD IgG titers reached peak levels after immunization with the second vaccine dose in the whole study population. However, in pre-vaccine SARS-CoV-2 infected subjects, an important increase in antibody titers was observed before the second vaccine dose and antibodies titers remained at high levels up to 6 months later. This is in agreement with previous studies that have demonstrated that BNT162b2 vaccines induce early robust antibody production in convalescent healthy subjects, suggesting a long-term anti-SARS-CoV-2 immunological memory of COVID-19 recovered patients [19,20,21]. On the contrary, in both post-vaccine SARS-CoV-2 infected and non-infected subjects the antibody titers showed a rapid decline up to 6 months post-second vaccine dose. Intriguingly, antibody titers in pre-vaccine infected individuals after the first vaccine dose were comparable to post-vaccine infected or naïve subjects who had received two doses, suggesting that infection may be equivalent to one vaccine dose.

Moreover, antibody concentrations decreased to the lowest level after approximately 9 months post-second vaccine dose in all study groups. Our finding is consistent with a recent observation from Ogrič et al., who also reported the greatest decrease in antibody titer at the same time point [22]. Accumulating evidence shows that a significant decrease in antibodies is observed approximately 70–90 days after the second vaccination [10,23]. Interestingly, in a large-scale study, antibody titers after two doses of mRNA vaccine reduced by up to 38% in each subsequent month, whereas in previously infected subjects they fell by less than 5% monthly [24]. Our study adds significant information regarding the long-term evolution of waning immunity against SARS-CoV-2 after vaccination and/or infection. Taking into consideration the timing of the antibody concentration nadir after the second vaccine dose, booster revaccination no later than 9 months post-second vaccine for susceptible groups could be a reasonable recommendation for future vaccination strategies. 

In addition, two months after the third dose, increased antibody titers were observed in the three study groups. However, the third dose led to even higher antibody titers compared to those measured post second dose, which is in line with previous studies [22,25]. In particular, we found that post-vaccine SARS-CoV-2 infected subjects, as well as naïve individuals, who presented a lower humoral response post-second dose, had approximately 2–3-fold higher anti-S-RBD IgG antibodies post-third vaccine dose. As in a previous study [22], this finding can be partially explained by the observation that the antibody response increases after each booster dose and/or natural infection and eventually reaches a plateau. This plateau in some people may be reached after the second dose, and a third dose does not lead to a noticeable increase, while in other people the third dose results in a stronger antibody response. Moreover, data from experimental mouse models have shown that that high levels of immunoglobulin may attenuate recall responses [26]. 

Of note, pre- and post-vaccine infected healthcare workers presented even higher antibody levels 6 months after a third dose, whereas naïve individuals showed a significant decrease in antibody titers at the same time point. More specifically, post-vaccine infected subjects exhibited higher antibody titers at 6 months post-third dose, compared with pre-vaccine infected subjects and naïve individuals. These results may be explained by the fact that the majority of post-vaccine SARS-CoV-2 infections and reinfections (73% and 54%, respectively) were observed after the third dose. Hence, natural infection may contribute significantly to a further antibody response. 

Several studies hitherto have shown an overall advantage for younger individuals in immune response and a negative effect of older age on immunogenicity after vaccination [27,28]. Interestingly, antibody levels appear to fall for each 10-year increase in age [29]. It is well known that biological aging causes a progressive immunosenescence leading to decreased cellular and humoral immune responses. Actually, age-related changes in both the quantity and quality of immune cells and soluble mediators affect both the innate and adaptive immune responses. Thus, older individuals present not only an increased susceptibility to infectious diseases but also a reduced response to vaccination [30]. Indeed, in our study, age-related differences were observed in the kinetics of antibodies at 2 months after the second dose and at 2 months after the third dose; specifically, as age increases, humoral immune response decreases. On the other hand, few studies have been conducted to investigate possible differences in antibody kinetics between men and women. Immune response following BNT162b2 vaccine has been demonstrated to be age- and gender-dependent, as it tended to be more robust in younger ages and in female octogenarians [31]. Anastassopoulou et al. found significant differences between males and females, showing a clear predominance in antibody titers for females in most age categories, as well as between individuals in the youngest people (21–31 years) and in the older age groups one and three months after vaccination with two doses of the BNT162b2 mRNA vaccine [32]. Similar findings were observed by population antibody surveillance following COVID-19 vaccination in England for both BNT162b2 and ChAdOx1 vaccines [33]. Moreover, in a study involving 358 subjects who received two doses of Coronavac, an inactivated whole-virion vaccine, the frequency of anti-SARS-CoV-2 total antibodies was significantly increased in females, whereas a high prevalence of anti-SARS-CoV-2 neutralizing antibodies was observed among young adults [34]. On the contrary, in a study including 268 healthcare workers who received two doses of the BNT162b2 vaccine, not statistically significant differences were observed in immune responses between the genders in terms of total and neutralizing anti-S-RBD IgG antibodies [35]. In accordance with the abovementioned study, in the current study we did not find significant sex-related differences regarding the kinetics of antibodies. Accumulating data shows that females compared to males present increased inflammatory, antiviral and humoral immune responses to COVID-19 [36]. Interestingly, antibody responses to a broad range of vaccines are remarkably higher in females than males. Sex-related differences in immune responses are observed even in advancing age groups, suggesting that circulating levels of sex steroid hormones are not the only reason for these differences [37]. Genetic or other factors may be responsible for female immunological advantages in antibody responses to vaccination. Investigating specific factors involved in sex-based differences in protective immune responses may allow for a better understanding of vaccine immunogenicity and the future development of more effective vaccines.

Furthermore, in our study, subjects who were infected after receiving at least one vaccine dose had fewer hospital admissions and a lower rate of reinfections than those who were infected before vaccination, confirming the protective effectiveness of vaccines as previously described in clinical studies [1,2,3,4]. In a previous observational study, we have shown that SARS-CoV-2 antibody responses did not predict clinical outcomes in hospitalized patients with moderate to severe COVID-19 [38]. Notably, in a recent systematic review and meta-regression analysis, the efficacy of vaccines declined within months against reinfection, but remained high for protection against hospitalization or severe disease due to the SARS-CoV-2 Omicron variant. Actually, subjects with hybrid immunity, which develops by a combination of natural SARS-CoV-2 infection and vaccination, had the highest level of protection compared to non-infected individuals [39]. 

Finally, in our study younger subjects had higher reinfection rates than older subjects. Moreover, in pre-vaccine SARS-CoV-2 infected subjects who were younger, the time from the second dose to reinfection was shorter compared with post-vaccine infected persons who were older. Among young persons, low compliance with COVID-19-related public health recommendations for social distancing, wearing of face masks, hand hygiene, and other protective measures might elevate their level of exposure compared with older subjects [40]. Moreover, in a prospective cohort study including healthy young adults aged 18–20 years, the risk of subsequent SARS-CoV-2 infection in previous infected persons was related to lower IgG antibody levels and absent or lower neutralizing antibody activity. Thus, although humoral immune responses induced by first SARS-CoV-2 infection are largely protective, they do not guarantee effective immunity against reinfection [41].

### Study Limitations

The present study has certain limitations. Firstly, our study was an observational study with retrospective data collection. Hence, the results of our study cannot be generalized to the general population. Furthermore, there was no availability of antibody titers before vaccination or serological testing in all participants in each sampling time. Additionally, sampling time points for antibody measurement differed among SARS-CoV-2 infected subjects due to modification of vaccine schedule by the infection. Another limitation of the current study was that we did not evaluate the neutralizing properties of antibodies; therefore, our findings cannot be considered to provide sufficient evidence of the degree to which vaccination protects subjects from SARS-CoV-2 infection. Finally, although the majority of subjects were vaccinated with mRNA-based vaccines, a percentage of 4.6% received a viral vector-based vaccine with a different dosing schedule, which may potentially affect antibody kinetics. In addition, we cannot estimate whether different vaccine type induce different levels of humoral immune response.

## 5. Conclusions

In conclusion, pre-vaccine infection leads to the rapid development of high antibody titers and a slower decline in a large cohort of Greek healthcare workers, suggesting that a modified vaccination schedule may be necessary in convalescing subjects. Moreover, vaccination is strongly associated with fewer hospitalizations and fewer reinfections, indicating the effectiveness of vaccines. Therefore, our study can improve knowledge about the kinetics and durability of antibody responses after vaccination against SARS-CoV-2 infection and can contribute to better protection not only among healthcare workers but also in groups that are vulnerable to severe COVID-19.

## Figures and Tables

**Figure 1 jpm-13-00910-f001:**
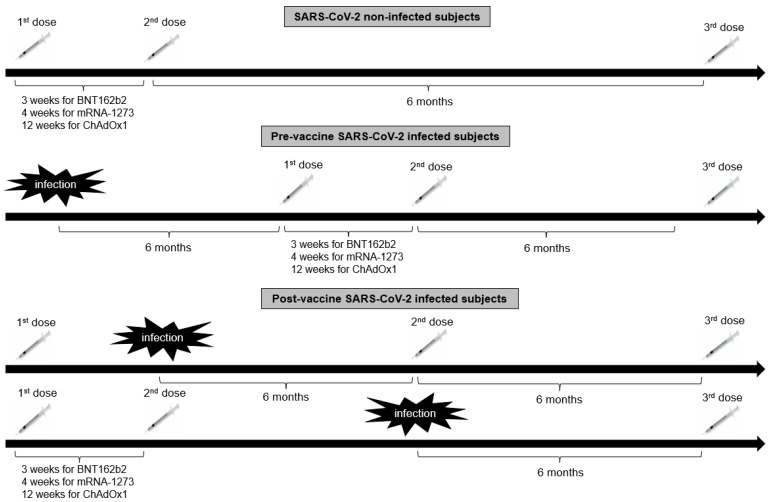
The timeline of vaccine administration in relation to history of SARS-CoV-2 infection in the study population.

**Figure 2 jpm-13-00910-f002:**
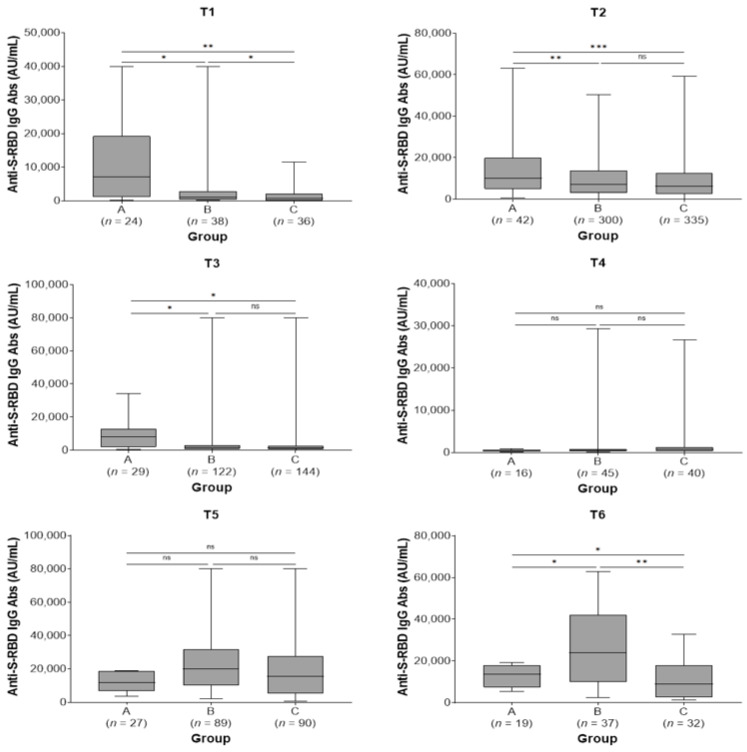
Anti-spike-receptor binding domain immunoglobulin G (anti-S-RBD IgG) antibody titers in the three study groups by sampling period. A, pre-vaccine SARS-CoV-2 infected subjects; B, post-vaccine SARS-CoV-2 infected subjects; C, SARS-CoV-2 non-infected subjects; T1, pre-second vaccine dose; T2, 2 months post-second vaccine dose; T3, 6 months post-second vaccine dose; T4, 9 months post-second vaccine dose (pre-third dose); T5, 2 months post-third vaccine dose; T6, 6 months post-third vaccine dose. ns, not significant (*p* > 0.05); * *p* < 0.05, ** *p* < 0.01, *** *p* < 0.001.

**Figure 3 jpm-13-00910-f003:**
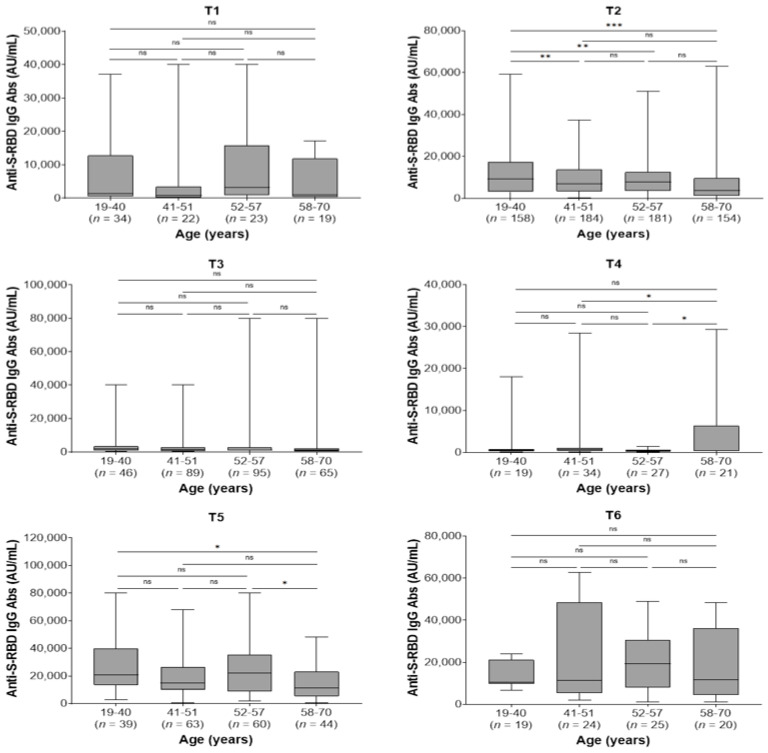
Anti-spike-receptor binding domain immunoglobulin G (anti-S-RBD IgG) antibody titers in different age groups by sampling period. T1, pre-second vaccine dose; T2, 2 months post-second vaccine dose; T3, 6 months post-second vaccine dose; T4, 9 months post-second vaccine dose (pre-third dose); T5, 2 months post-third vaccine dose; T6, 6 months post-third vaccine dose. ns, not significant (*p* > 0.05); * *p* < 0.05, ** *p* < 0.01, *** *p* < 0.001.

**Table 1 jpm-13-00910-t001:** Demographics and clinical characteristics of the study population.

	All Subjects	Pre-Vaccine SARS-CoV-2 Infected Subjects	Post-Vaccine SARS-CoV-2 Infected Subjects	SARS-CoV-2 Non-Infected Subjects
	*n* = 803	*n* = 100	*n* = 335	*n* = 368
Age, years	51 (40–57)	46.5 (30–55) *	49 (39–56) ^‡‡^	53 (43–58) ^‡‡‡^
Sex (female), *n* (%)	540 (67.2)	62 (62)	238 (71)	240 (65.2)
Vaccine, *n* (%)				
BNT162b2 (Pfizer-BioNTech)	758 (94.4)	91 (91)	319 (95.2)	348 (94.6)
mRNA-1273 (Moderna)	8 (1)	2 (2)	2 (0.6)	4 (1.1)
ChAdOx1 (Oxford-AstraZeneca)	14 (1.7)	1 (1)	4 (1.2)	9 (2.5)
Ad26.COV2.S (Janssen)	23 (2.9)	6 (6)	10 (3)	7 (1.9) ***
Hospitalization, *n* (%)	13 (1.6)	8 (8) ^†^	5 (1.5)	-
Intubation, *n* (%)	1 (0.1)	1 (1)	0 (0)	-
SARS-CoV-2 reinfection, *n* (%)	39 (4.9)	28 (28) ^‡^	11 (3.3)	-
Time from 2nd vaccine dose to first SARS-CoV-2 infection, months		-	11.9 (8.2–14)	-
Time from 2nd vaccine dose to SARS-CoV-2 reinfection, months	11.1 (6–13.1)	8.9 (4.6–13.4) *	11.4 (11–12)	-
Co-morbidities, *n* (%)				
Obesity	110 (13.7)	14 (14)	49 (14.6)	47 (12.8)
Hypertension	112 (13.9)	15 (15)	55 (16.4)	42 (11.4)
Dyslipidemia	142 (17.7)	18 (18)	58 (17.3)	66 (18)
Diabetes mellitus	41 (5.1)	4 (4)	21 (6.3)	16 (4.3)
CAD	14 (1.7)	5 (5) *	5 (1.5)	4 (1.1) ***
COPD	4 (0.5)	0 (0)	3 (0.9)	1 (0.3)
Cancer	16 (2)	2 (2)	4 (1.2)	10 (2.7)
Immunosuppression	7 (0.9)	3 (3) *	2 (0.6)	2 (0.5) ***
Number of co-morbidities, *n* (%)				
None	542 (67.5)	65 (65)	225 (67.1)	252 (68.5)
1	143 (17.8)	19 (19)	55 (16.4)	69 (18.8)
2	64 (8)	9 (9)	28 (8.4)	27 (7.3)
≥3	54 (6.7)	7 (7)	27 (8.1)	20 (5.4)
Medications, *n* (%)				
ACEi/ARBs	90 (11.2)	12 (12)	45 (13.4)	33 (9)
CCB	69 (8.6)	8 (8)	36 (10.7)	25 (6.8)
β-Blockers	39 (4.9)	5 (5)	23 (6.8) **	11 (3)
Diuretics	31 (3.9)	5 (5)	12 (3.6)	14 (3.8)
Antiplatelets	26 (3.2)	6 (6)	10 (3)	10 (2.7)
Anticoagulants	10 (1.2)	0 (0)	5 (1.5)	5 (1.4)
Statins	139 (17.3)	18 (18)	58 (17.3)	63 (17.2)
Fibrates	3 (0.4)	1 (1)	2 (0.6)	0 (0)
Antidiabetics	35 (4.4)	4 (4)	17 (5.1)	14 (3.8)
Insulin	8 (1)	0 (0)	2 (0.6)	6 (1.6)
Corticosteroids	10 (1.2)	2 (2)	2 (0.6)	6 (1.6)

Data are presented as median values (first quartile–third quartile) or number (%). CAD, coronary artery disease; COPD, chronic obstructive pulmonary disease; ACEi, angiotensin-converting enzyme inhibitors; ARBs, angiotensin receptor blockers; CCB, calcium channel blockers. * *p* < 0.05, ^†^
*p* = 0.001, ^‡^
*p* < 0.001 for comparisons of pre-vaccine SARS-CoV-2 infected subjects vs. post-vaccine SARS-CoV-2 infected subjects. ** *p* < 0.05, ^‡‡^
*p* < 0.001 for comparisons of post-vaccine SARS-CoV-2 infected subjects vs. SARS-CoV-2 non-infected subjects. *** *p* < 0.05, ^‡‡‡^
*p* < 0.001 for comparisons of SARS-CoV-2 non-infected subjects vs. pre-vaccine SARS-CoV-2 infected subjects.

**Table 2 jpm-13-00910-t002:** The timing of SARS-CoV-2 infection and reinfection in the study population.

Pre-vaccine SARS-CoV-2 infection (*n* = 100)	
	***n* (%)**
One infection pre-vaccination	72 (72)
Reinfection (pre- or post-vaccination)	28 (28)
**Post-vaccine SARS-CoV-2 infection (*n* = 335)**	
	***n* (%)**
Between first and second vaccine dose	10 (3)
Between second and third vaccine dose	82 (24)
Post-third vaccine dose	243 (73)
**Total SARS-CoV-2 reinfection (*n* = 39)**	
	***n* (%)**
Prior to vaccination	4 (10)
Between first and second vaccine dose	5 (13)
Between second and third vaccine dose	9 (23)
Post-third vaccine dose	21 (54)

**Table 3 jpm-13-00910-t003:** Antibody titer levels of the study groups.

	All Subjects	Pre-Vaccine SARS-CoV-2 Infected Subjects	Post-Vaccine SARS-CoV-2 Infected Subjects	SARS-CoV-2 Non-Infected Subjects
	*n* = 803	*n* = 100	*n* = 335	*n* = 368
Anti-S-RBD IgG Abs (AU/mL) pre-second vaccine dose (2 months post-first dose; T1)	1285 (419–11,688) (*n* = 98)	7181 (1120–19,257) * (*n* = 24)	1136 (330–2859) ** (*n* = 38)	623 (113–2190) ^†††^ (*n* = 36)
Anti-S-RBD IgG Abs (AU/mL) at 2 months post-second vaccine dose (T2)	6934 (2601–13,091) (*n* = 677)	10,134 (4852–20,025) ^†^ (*n* = 42)	7144 (2916–13,848) (*n* = 300)	6169 (2399–12,554) ^‡^ (*n* = 335)
Anti-S-RBD IgG Abs (AU/mL) at 6 months post-second vaccine dose (T3)	1370 (664–2864) (*n* = 295)	7876 (1792–12,944) * (*n* = 29)	1426 (661–2968) (*n* = 122)	1266 (613–2690) *** (*n* = 144)
Anti-S-RBD IgG Abs (AU/mL) at 9 months post-second vaccine dose (pre-third dose; T4)	560 (273–1000) (*n* = 101)	432 (259–701) (*n* = 16)	512 (272–833) (*n* = 45)	649 (279–1257) (*n* = 40)
Anti-S-RBD IgG Abs (AU/mL) at 2 months post-third vaccine dose (T5)	17,203 (8607–28,252) (*n* = 206)	11,887 (6655–18,841) (*n* = 27)	19,876 (10,077–31,755) (*n* = 89)	15,524 (5385–27,710) (*n* = 90)
Anti-S-RBD IgG Abs (AU/mL) at 6 months post-third vaccine dose (T6)	14,226 (5894–28,137) (*n* = 88)	13,590 (7302–17,994) * (*n* = 19)	23,848 (9829–42,075) ^††^ (*n* = 37)	8882 (2474–17,819) *** (*n* = 32)

Data are presented as median values (first quartile–third quartile). Anti-S-RBD IgG Abs, anti-spike-receptor binding domain immunoglobulin G antibodies. * *p* < 0.05, ^†^
*p* < 0.01 for comparisons of pre-vaccine SARS-CoV-2 infected subjects vs. post-vaccine SARS-CoV-2 infected subjects. ** *p* < 0.05, ^††^
*p* < 0.01 for comparisons of post-vaccine SARS-CoV-2 infected subjects vs. SARS-CoV-2 non-infected subjects. *** *p* < 0.05, ^†††^
*p* < 0.01, ^‡^
*p* < 0.001 for comparisons of SARS-CoV-2 non-infected subjects vs. pre-vaccine SARS-CoV-2 infected subjects.

## Data Availability

The datasets generated and/or analyzed during the current study are not publicly available due to information that could compromise the privacy of research participants but are available from the corresponding author upon reasonable request. All of other data is contained within the article.

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
