# Peer review of "Evaluation of Antibody Kinetics Following COVID-19 Vaccination in Greek SARS-CoV-2 Infected and Naïve Healthcare Workers"

_jpm, 2023, doi:10.3390/jpm13060910_

Round 1

Reviewer 2 Report

The comments to the authors are attached.

Reviewer 3 Report

The manuscript of Plavids et al. is the first to investigate the antibody kinetics after mRNA vaccination for Covid-19 in Greece. It was shown that anti-S-RBD-IgG titlers reached peak levels after immunization with the second vaccine dose. Furthermore, the titlers were higher in those healthy subjects in the recovery from a previous infection, suggesting that the infection can be equivalent to one vaccine dose.

The manuscript is very well written and the methods are adequately described. 

Minor review

The authors should change the statistical significance symbol "?" for "*" in all figures.  

Suggestion

In the discussion, regarding the differences in IgG anti-S-RBD between sex and age, maybe the authors could cite the study below, about the assessment of humoral immune response after the CoronaVac vaccine.

Bichara CDA,  et al. Assessment of Anti-SARS-CoV-2 Antibodies Post-Coronavac Vaccination in the Amazon Region of Brazil. Vaccines (Basel). 2021 Oct 12;9(10):1169. doi: 10.3390/vaccines9101169.
